# Sustainability of Local Labour Market in South Africa: The Implications of Imports Competition from China

**Muftah Faraj \* and Murad Bein** 

Faculty of Economics and Administrative Sciences, Cyprus International University, North Cyprus, Mersin 10, Turkey; mbein@ciu.edu.tr

\* Correspondence: 20165681@student.ciu.edu.tr

**Abstract:** Using South African manufacturing and non-manufacturing industry employment, imports from China, growth in manufacturing and non-manufacturing industry, and workers' earning data, we examined the impact of imports from China, growth in manufacturing and non-manufacturing industry on manufacturing and non-manufacturing industry employment and workers' earnings. This study employed a Bayer and Hanck cointegration test to examine the cointegration among the variables, which found the existence of cointegration. In addition, the ARDL approach was employed to ascertain the long-run effect of the import from China, growth in manufacturing and non-manufacturing industry on the manufacturing and non-manufacturing industry employment and workers' earning, while "Fully Modified Ordinary Least Square (FMOLS)", "Dynamic Ordinary Least Square (DOLS)", and "Canonical Cointegrated Regression (CRR)" estimators were employed for robustness. We found a negative long-run effect of imports from China on manufacturing sectors' employment and workers' earnings, while a positive of its effect was found on service industry employment. Moreover, growth in the manufacturing industry was found to have a positive long-run effect on manufacturing industry employment and workers' earnings, while it has a negative long-run effect on service industry employment. As for the growth in the service industry, it was demonstrated to have a negative and positive long-run effect on manufacturing industry employment and non-manufacturing industry employment.

**Keywords:** sustainability; labour market; import shock; international trade; South Africa

## 1. Introduction

It is not in doubt that China has been growing rapidly in strength and rising internationally in the recent years [1]. This development has posed a threat to the point that some countries have ceaselessly put in place plans to check China, with the aim of maintaining their regional control and partial balance [1,2]. Meanwhile, the deepening of globalization has practically made it impossible for any country to remain independent, especially when confronted with difficult economic situations and challenges. However, Hernandez [2] opined that there is a significant influence of international trade on the labour market. This position was in congruence with some previous studies [3,4], which posited that income, level of employment and other labour market outcomes are being influenced by international trade. Owing to this fact, Baldwin and Okubo [5] argued that the impact of international trade on labour markets have manifested in countries of all income levels. The consequences are evident in the excessive and increasing income inequality as espoused by ILO [6–8] and UNCTAD [9], which is within the context of "hyperglobalization" [10].

Moreover, Jiang et al. [11] opined that there is an increase in tariff barriers, the frequent imposition of retaliatory tariffs, as well as the intensification of international frictions between countries. The study further opined that while there is evidence of shrinking free trade and strengthening of protectionists, the trend poses a threat to the sustainability of the international trade order. This trend has intensified the concern of various stakeholders

who observed that the rapid growth of trade low- and high-income economies calls for concern, especially in refence to its influence on the domestic labour market in low-income countries [12–23]. In the last few decades, China's trade with other countries has dramatically changed the patterns of world trade, given a cause for policymakers to have a rethink on the adjustment of domestic and their implications [11]. In reference to UNCTAD [24], there is an increase of China's share of world exports from below 2% to 14.7% between 1998 and 2020.

Furthermore, records show that an all-time high in the trade between China and Africa was reached in 2021 with an estimated value of USD 254 billion [24]. About USD 148 billion worth of goods was exported by China to Africa, which shows an increase of about 29.9% on 2020 (see Figure 1), while China receiving USD 106 billion worth of goods in import from the continent—a rise of 43.7% from the previous year [24]. In view of the large size of the China–Africa trade, there is possibility of the trade having influence on the African countries labour market, especially in the manufacturing sector. Though, previous studies pointed out that imports from China could have disruptive impacts on labour markets in developed countries, and in particular, pose a threat to low-skilled workers in industries competing with imports from low-cost countries.

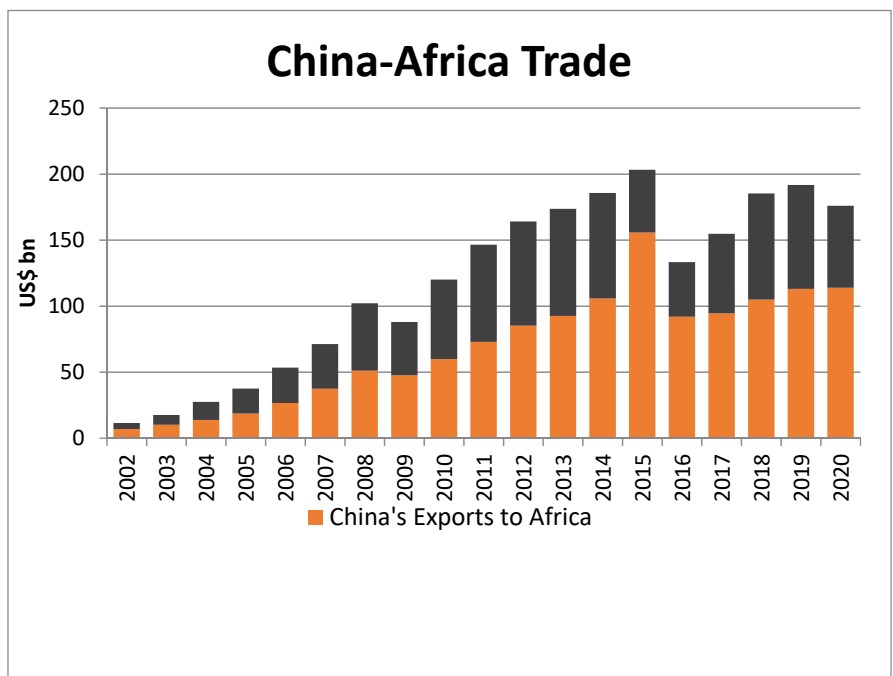

**Figure 1.** China–Africa trade (2002–2020). Source: UN Comtrade.

However, even though the attention of scholars has over time shifted to investigating the relationship between trade and the labour market [25–32], to the best of our knowledge, none of the previous studies have explored the employment effect of import shock from China on the developing countries' labour markets, especially South Africa's labour market. Although, previous early studies argued that the increase in wage inequality observed in many countries could be attributed to skilled-biased technological changes and not necessarily trade [15–19]. Meanwhile, as trade increased between developed and developing countries, especially China, several studies have found the influence of trade shocks on labour markets [12–14,19,21].

South Africa is an interesting country to study the influence of trade shocks in the labour market as the record shows that the country is the second importer country from China, coming behind Nigeria (see Figure 2).

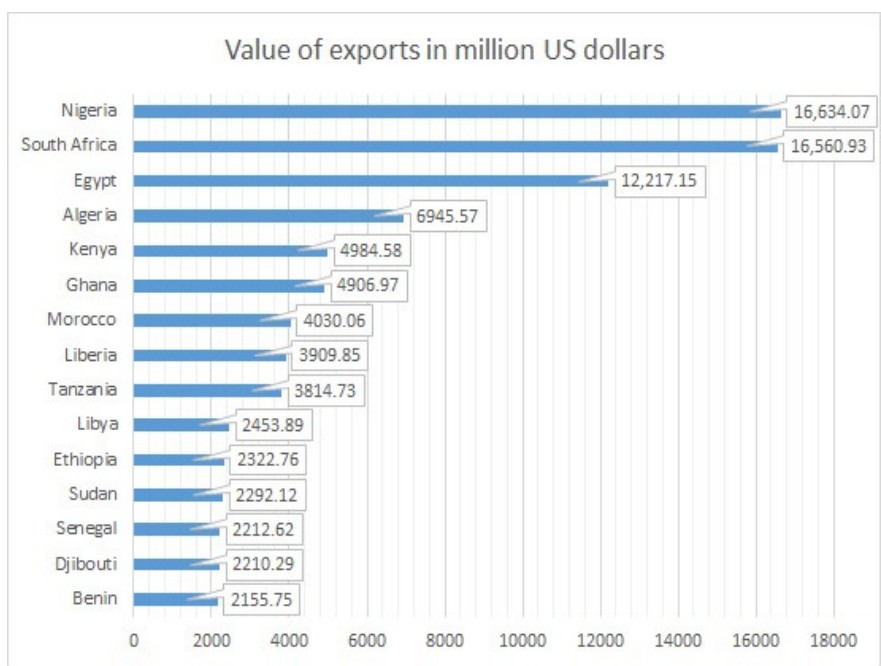

**Figure 2.** Value of Chinese exports to Africa in 2019, by country (in million U.S. dollars). Source: Statista 2022.

Meanwhile, Ranchhod and Daniels [33] observed that the labour market in South Africa has been revealed by several studies to be the primary institution for determining some socio-economic welfare measures. Moreover, the country is considered to be one of the most unequal countries in the world and the main contributor of economic inequality is also mediated through access to formal sector employment [33]. Moreover, Statistics South Africa [34] shows that between 2013 and 2018, the South Africa employment rate increased by 1.5 million from 14.9 million to 16.4 million. Additionally, during the same period, the unemployment rate and absorption rate increased by 2.4% and 0.5%, respectively. Meanwhile, the country economic growth declined from a high of 2.5% in 2013 to 0.8% in 2018.

In this study, we investigate how the increased exposure to imports from China would affect the sustainability of the local labour market in South Africa, using data from 1992–2020. This study's point of departure from previous studies is the increasing attention of researchers towards investigating the regional dimensions of trade shocks with taking cognizance of the fact that regional differences in the production and employment structure within countries tend to make some regions more vulnerable to trade shocks. Our study is based on the Autor et al. [25] approach that investigates the effect of increased exposure to imports from China from numerous labour market outcomes in US commuting zones. The study found that about 20% of the employment share of manufacturing decrease in the US between 1990 to 2007 was a result of the increase in imports from China during the period. These findings from Autor et al. [25] corroborate the position of Bernard et al. [35], who argued that US manufacturing firm survival and growth are negatively associated with industry-level exposure to import competition from low-wage countries. This is an indication that the sustainability of the local labour market could be associated with import competition from a country like China. The study of Citino and Linarello [15] opined that in the labour market under new technology and trade, low-skilled workers are known to be susceptible, and South Africa is addressing this challenge through education policy intervention, which is slowly paying off as the low-skilled are gaining more skills and an increased share of skilled labour income.

In addition, according to Citino and Linarello [15], the import from low-income countries, especially China has raised the concerns about the income of wage earners in the destination countries. This has been empirically investigated by some studies [26,36].

However, they are not exhaustive, especially how the import shock affects the wage-earning growth. Meanwhile, similar study conducted by Jiang et al. [11] on the Sweden labour market revealed an insignificant impact of increasing exposure of the country to import from China on the manufacturing and non-manufacturing employment growth of the country. However, the study found a significant impact of the trade shocks from China on the wage earning at the median level. Therefore, this study will investigate the impact of trade shock from China on the wage earnings in South Africa. This investigation would contribute to the growing literature on the effect of trade shocks on employment and wage-earning sustainability, especially in developing countries. Thus, our contribution to the active literature is on the effect of increasing import competition from China on the sustainability of labour market in a developing country.

The novelty of our study lies in the existing gap in the literature, which indicates the absence of studies that explored the impact of increasing exposure to imports from China on the South African manufacturing and non-manufacturing employment growth, as well as the trade shock impact on the wage earning.

The remainder of this paper is structured as follows. The relevant previous studies that relate to the international trade and labour markets were reviewed and presented in Section 2, while the explanation on the dataset and the method employed for the analysis were presented in Section 3. In the subsequent section, the empirical results were interpreted, while the discussions of the empirical results and conclusion followed in Section 5.

## 2. Literature Review

The studies of Wood [37] and Autor et al. [25,38] presented a theoretical foundation that motivates the measurement of the impact of international trade on a local labour market. The studies opined that an exogenous shock like an international trade would have an impact on the regional economies in some countries through two channels: import-demand and export supply shocks. According to Jiang et al. [11], in an industry that is exposed to trade, especially in high-income, the demand for labour in such an industry is often reduced owing to the "export-supply shock" from low-income countries. In this scenario, there is expectation of flow of the labour force from the industry exposed to international trade to other industries that are not exposed to this shock or drift into unemployment. As for the "import-demand shock", there is expectation of an increase in import demand to high-income countries' regional wages and employment on the traded industry. In this case, the labour force is expected to drift from the industry not engaged in international trade to others that are involved in it or opt for unemployment. The studies also espoused that a large negative impact like unemployment on regional labour markets, especially those concentrated in the manufacturing industry, as well as cumulative earnings of low-skilled workers is expected with the increasing import competition from China. However, Wood [37] and Autor et al. [25,38] opined that the effect size is weak among the high-skilled workers. The findings from these studies were corroborated with the positions of Donoso, Martin, and Minondo [39] and Bilici [40], who conducted similar studies in Spain and France, respectively, and concluded that the Chinese import competition mostly impacted on the manufacturing sectors' labour force.

Meanwhile, there was mixed results from other studies [26,40,41], which were conducted within the context of Northern European countries. These studies concluded that there is a small impact of import trade on manufacturing employment in these countries. In reference to Bilici [40], who studied the UK manufacturing labour market employment and wages, the study revealed a little impact of Chinese import shock on the labour market. As for Dauth et al. [41], a negative effect of trade with Eastern European countries on the local labour market was found instead of China's import effect in the context of Eastern European countries. Though, Balsvik et al. [26] found a negative influence of increasing import exposure from China on Norway's manufacturing employment, but the impact is quantitatively lower that what Autor et al. [25] identified in their study. The negative

effect of China's import exposure on earnings could not be established by Balsvik et al. [26], which was argued to be the centralized wage bargaining system of Norway. A similar study was conducted by Utar [42] using Danish employer–employee data from 1995–2007. The study established a negative employment effect in the Danish textile and clothing industry owing to the increasing import of Chinese textile products. In addition, the study demonstrates that there is a reaction from the firms to the Chinese competition through their diversion to other products aside from the Chinese competitive products and enabled them to become more skill-oriented. The study of Baziki [43] is tilted towards the effect of China import shock on earnings. The study demonstrates that the penetration of Chinese imports significantly triggers the earnings of high-skilled workers, which could be wooing to the upgrade of their firm's production technology, in their response to the competition from low-income countries, which necessitate the changes in the demand for skilled labour. However, a significant negative effect of Chinese import shock on the earnings of low-skilled workers could not be established in the study of Baziki [43].

The case of Mexico was investigated in a recent study by Blyde et al. [13]. The study examined the response of Mexico's labour market to the increase in Chinese import competition and finds a more severe negative employment effect on manufacturing employees than non-manufacturing employees. This indicates that the low-skilled employees were more severely influenced. Another recent study by Liang [21] examined the trade shocks' impact on US manufacturing employment, and the result shows a comparison between the job creation obtained from the exports to different markets and the job destruction attributed to the import competition from China. Similarly, it was demonstrated in the study of Citino and Linarello [15] that those long-term losses in terms of lower earnings or more discontinuous careers could not be established for employee initially employed in a more exposed industry. Still on the US labour market, Lang et al. [20] examined the Chinese import exposure on the market and discovered that on average, all the physical, mental, and general health workers in the US labour market are exposed to greater import competition. This finding was similar to the study of Branstetter et al. [14], who conducted a study in the context of Portugal, examining the effect of Chinese import competition on firm-level labour market outcomes in the country. The study demonstrates a significant employment decline in firms with more exposure to Chinese competition in European export markets, but minimal impact of direct competition in Portugal.

A similar result was also established in the study of Kazunobu et al. [44] for Japan's labour market. Meanwhile, a recent study by Jiang et al. [11] in the context of Sweden reveals that, except for the transportation sector, the manufacturing and non-manufacturing employment growth were not statistically influenced by the increasing import exposure to China. As for the effect on the earnings, the study established that earnings growth of low-wage employees in the manufacturing sector was not significantly affected, but the wage earnings of the media level or above were positively influenced by the import shocks from China. The findings from the study of Jiang et al. [11] is in contrast to the position of Blanco et al. [12] who conducted similar study in Australia and revealed a significant negative effect of imports from China on the manufacturing employment in Australia. In addition, Jiang et al. [11] was at variance to the position of Taniguchi [31], who established a positive effect of import from China on manufacturing employment in Japan.

From the literature reviewed, it is evident that there is paucity of studies that explored the labour markets in African countries, whereas the continent accommodates countries at developing stages. Thus, it becomes imperative to have a deeper understanding on the possible effect of exposure to increasing import from China on their local labour market. Hence, the aim of this study to fill the gap by testing the following hypotheses: whether imports from China influence (i) the manufacturing industry employment, (ii) non-manufacturing industry employment, (iii) wage earning of employees in South Africa, and (iv) to test the hypotheses of whether growth in the manufacturing and non-manufacturing industries influence the manufacturing and non-manufacturing industries' employment, as well as wage earning in South Africa's labour market.

## 3. Data and Methods

The trend of South African import from China between 1992 and 2020 as depicted in Figure 3 shows the continuous increase in the Chinese exports to the country; however, it was not until around 2017 when there was a sharp decline in the value, but it started rising again, but still not up to the all-time high recorded in 2015, before it starts decreasing again prior to the outbreak of COVID-19. For this reason, the analysis of the impact of Chinese exports on the South African labour market focuses on this period. This study covers the annual data to investigate the effect of import shock from China on the local labour market of South Africa in the period from 1992 to 2020, which was converted to quarterly data. The period covered in this study is subject to the data availability. The data for this study was sourced from different sources. China's exports to South Africa, which were sourced from UN Comtrade [45], were utilized for measuring the import shock from China (in million US$). As for the growth of manufacturing and non-manufacturing employment, manufacturing and non-manufacturing employment data were utilized as a proxy and were sourced from the International Labour Organization (ILO) Database. Similarly, the data for wage earning of South African workers was sourced from the ILO database and the data was measured in USD. The data, which was measured in nominal value, was converted to real wage value to eliminate any possible inflation on the wage earnings. The measures of growth manufacturing and non-manufacturing sector were proxy with the growth in both sectors as a percentage of annual growth and were sourced from World Bank Development Indicators. Meanwhile, owing to the short-time span availability of data for most of the variables, the series were converted from annual to quarterly data with the aim of ensuring the reliability of the estimates; and subsequently prior to the final analysis, the variables were transformed into natural logarithm form. As depicted in Table 1, the descriptive statistics of the variables indicate that growth of manufacturing industry (GMI) and non-manufacturing industry (GSI) ranges from −8% to about 12% and about 11% and 21%, respectively, with an average value of 3.18 and 16.28 within the period under study. Moreover, the import from China (IFC) as presented in Table 1 shows a value range between 0 and 16.936 billion USD with an average value of 7 billion USD. Additionally, the manufacturing (MIE) and non-manufacturing (SIE) employment ranges between 1.5 million and 2.16 million with an average employment of 1.8 million and 8.98 million employment, respectively. Meanwhile, the real wage earning in South Africa during the period under study shows a range of value between USD 148 and USD 1210 with an average value of USD 715.

**Table 1.** Descriptive statistics.

| Variable | Mean | Max. | Min. | Std. Dev. | Obs. |
|----------|------|------|------|-----------|------|
| GMI (%) | 3.177 | 11.835 | −8.003 | 2.844 | 116 |
| GSI (%) | 16.281 | 21.475 | 11.420 | 3.366 | 116 |
| IFC ($Mn) | 7070.66 | 16,936.7 | 0 | 6648.86 | 116 |
| MIE (Thousand) | 1795.87 | 2163.07 | 1510.73 | 142.35 | 116 |
| SIE (Thousand) | 8985.37 | 12,088.19 | 5550.33 | 2177.341 | 116 |
| WE (US$) | 715.22 | 1210.20 | 148.33 | 306.87 | 116 |

Note: GMI = growth manufacturing industry, GSI = growth service industry, IFC = import from China, MIE = manufacturing industry employment, SIE = service industry employment, WE = wage earning.

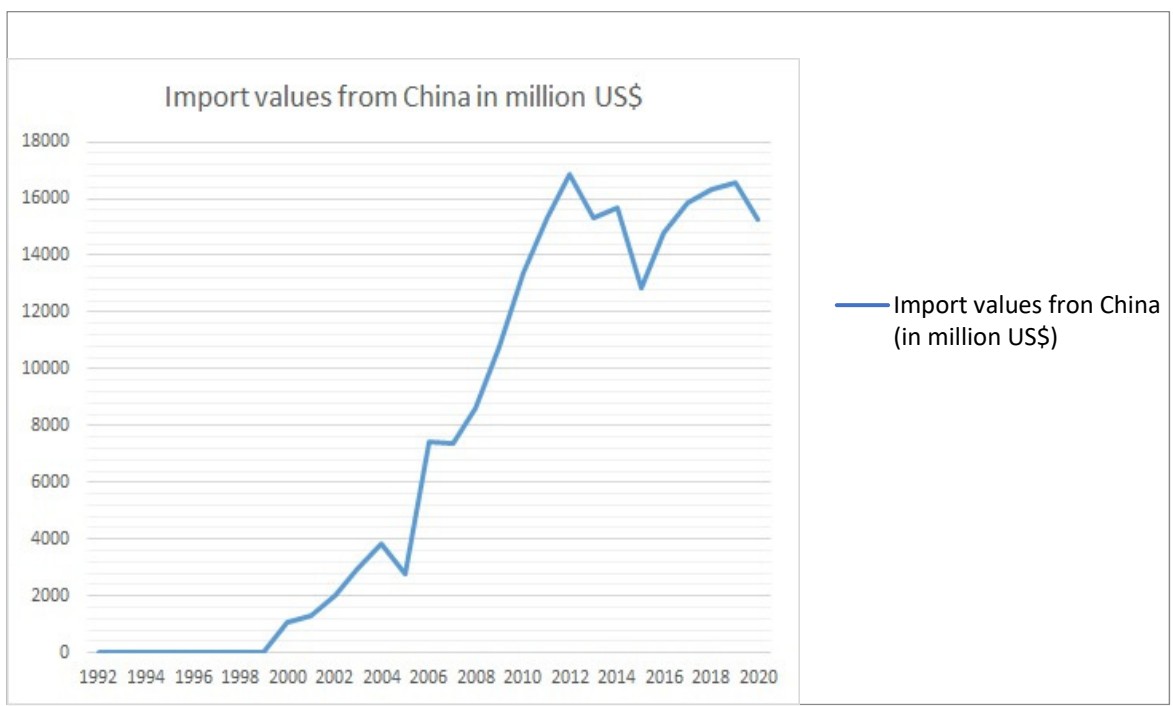

**Figure 3.** South African import values from China, 1992–2020 (in million).

## 4. Methods of Analysis

### 4.1. Unit Root Test

Unit root tests are important for performing time series analysis for examining the stationarity property of the variables. Hence, this study employed Augmented-Dickey Fuller test [46] and Phillips-Perron [47]. The inclusion of Ng and Perron test was owing to the consideration of the test being an effective modification of unit root tests. As for the PP, its robustness to a set of time-dependent serial correlations and heteroscedasticities justified its use in this analysis. Moreover, the Akaike information was utilized in this study to evaluate the lag length in the ADF test case, while the Newey–West Bartlett Kernel was utilized to select the bandwidth for the PP test. Applying the conventional integration approach, the null hypothesis suggests that the series is non-stationary, while it is stationary in the case of alternative hypothesis. For this study, intercept and deterministic trends of the integration of time series data have been used.

### 4.2. Cointegration Analysis

Over the past years, several cointegration methods have evolved. Engle and Granger [48] recommended a cointegration evaluation based on projected long-term regression model residuals. Several cointegration experiments were developed decades later, such as Johansen's system-based test [49], Boswijk's ECM-based F-test [50], and Banerje et al.'s ECM-based *t*-test [51]. The fact that these cointegration approaches have distinct theoretical bases and produce contradictory findings is a limitation. As a result, the effectiveness of ranking cointegration techniques is dependent on the value of nuisance estimators [52]. This research utilized the newly established cointegration test proposed by Bayer and Hanck [53] to verify the existence of the cointegration relationship between the variables in our models. A distinctive feature of this modern cointegration test is that it helps one to integrate multiple individual findings from the cointegration into a more decisive outcome. In this regard, Bayer and Hanck [53] recommend combining the individual tests' computed significance level (*p*-value) with the following Fisher's formulas:

$$EG - J = -2[ln(p_{EG}) + ln(p_J)] \tag{1}$$

$$EG - J - B - BDM = -2[ln(p_{EG}) + ln(p_J) + ln(p_B) + ln(p_{BDM})] \tag{2}$$

where, *p*-values of several distinct cointegration tests such as Engle-Granger [48]; Boswijik [50] and, Barnerje et al. [51] are shown by $p_{EG}$, $p_B$ and $p_{BDM}$, respectively. In Fisher statistics of estimated values when exceeding the critical values proposed by Bayer and Hanck [53], the null hypothesis of no cointegration can be rejected.

### 4.3. Long-Run Relationship Estimates

In this study, three models were developed to examine the long-run influence of IFC, GMI and GSI on the MIE (Model 1); the long-run influence of IFC, GMI, and GSI on SIE (Model 2); and the long-run influence of IFCm GMI, and GSI on WE (Model 2). The long-run estimates in these models were examined with ARDL model using the following equations:

$$
\begin{aligned}
\Delta lnMIE_t = \beta_0 &+ \sum_{i=1}^{k} \gamma_1 lnMIE_{t-i} \\
&+ \sum_{i=1}^{k} \gamma_2 lnIFC_{t-i} + \sum_{i=1}^{k} \gamma_3 \Delta lnGMI_{t-i} \\
&+ \sum_{i=1}^{k} \gamma_4 \Delta lnGSI_{t-i} + \sigma_1 lnMIE_{t-1} + \sigma_2 lnIFC_{t-1} + \sigma_3 lnGMI_{t-1} \\
&+ \gamma_4 lnGSI_{t-1} + \varepsilon_{1t}
\end{aligned}
\tag{3}
$$

$$
\begin{aligned}
\Delta lnSIE_t = \beta_0 &+ \sum_{i=1}^{k} \gamma_1 lnSIE_{t-i} \\
&+ \sum_{i=1}^{k} \gamma_2 lnIFC_{t-i} + \sum_{i=1}^{k} \gamma_3 \Delta lnGMI_{t-i} \\
&+ \sum_{i=1}^{k} \gamma_4 \Delta lnGSI_{t-i} + \sigma_1 lnMIE_{t-1} + \sigma_2 lnIFC_{t-1} + \sigma_3 lnGMI_{t-1} \\
&+ \gamma_4 lnGSI_{t-1} + \varepsilon_{1t}
\end{aligned}
\tag{4}
$$

$$
\begin{aligned}
\Delta lnWE_t = \beta_0 &+ \sum_{i=1}^{k} \gamma_1 lnWE_{t-i} \\
&+ \sum_{i=1}^{k} \gamma_2 lnIFC_{t-i} + \sum_{i=1}^{k} \gamma_3 \Delta lnGMI_{t-i} \\
&+ \sum_{i=1}^{k} \gamma_4 \Delta lnGSI_{t-i} + \sigma_1 lnMIE_{t-1} + \sigma_2 lnIFC_{t-1} + \sigma_3 lnGMI_{t-1} \\
&+ \gamma_4 lnGSI_{t-1} + \varepsilon_{1t}
\end{aligned}
\tag{5}
$$

where the first difference is denoted by $\Delta$, *lnMIE*, *lnSIE*, *lnIFC*, *lnGMI*, *lnGSI* and *lnWE* are the tested variables; the lag optimal is represented with k, while $\varepsilon_{1t}$ denoted the error term of the investigated models. Moreover, the error correction of term (ECT) for Equations (3)–(5) are formulated in Equations (6)–(8):

$$
\begin{aligned}
\Delta lnMIE_t = \beta_0 &+ \sum_{i=1}^{k} \beta_1 lnMIE_{t-i} \\
&+ \sum_{i=1}^{k} \beta_2 lnIFC_{t-i} + \sum_{i=1}^{k} \beta_3 \Delta lnGMI_{t-i} \\
&+ \sum_{i=1}^{k} \beta_4 \Delta lnGSI_{t-i} + ECT_{t-1}\, u_t
\end{aligned}
\tag{6}
$$

$$
\begin{aligned}
\Delta lnSIE_t = \beta_0 &+ \sum_{i=1}^{k} \beta_1 lnSIE_{t-i} \\
&+ \sum_{i=1}^{k} \beta_2 lnIFC_{t-i} + \sum_{i=1}^{k} \beta_3 \Delta lnGMI_{t-i} \\
&+ \sum_{i=1}^{k} \beta_4 \Delta lnGSI_{t-i} + ECT_{t-1}\, u_t
\end{aligned}
\tag{7}
$$

$$\Delta lnWE_t = \beta_0 + \sum_{i=1}^{k} \beta_1 lnWE_{t-i}$$
$$+ \sum_{i=1}^{k} \beta_2 lnIFC_{t-i} + \sum_{i=1}^{k} \beta_3 \Delta lnGMI_{t-i} \tag{8}$$
$$+ \sum_{i=1}^{k} \beta_4 \Delta lnGSI_{t-i} + ECT_{t-1} u_t$$

where ECT denoted an error correction term, and the negative sign of the ECT coefficient and its significance approves the adjustment speed in the short and long levels.

In order to ensure the reliability of the estimates from the ARDL estimator, and also to avoid functional misspecification owing to the volatility of the series, diagnostic tests for the parameters are required [54,55]. Thus, the Breusch–Pagan Godfrey test for serial correlation ($LM^{test}$) and heteroscedasticity ($Heterscedasticity^{test}$), normality and stability test using the CUSUM and CUSUM square were conducted.

Subsequently, the Fully Modified Ordinary Least Squares (FMOLS) estimator developed by Hansen and Phillips [56] and the Dynamic Ordinary Least Square (DOLS) advocated by Stock and Watson [57], which have several advantages over OLS estimators [54,55] were employed. FMOLS applies on a semi-parametric approach for long-run parameter estimation [58–60]. FMOLS provides consistent estimator parameters [54,55,61,62] and transform data and parameters [63] in the case of small samples and succession over endogeneity or serial measurement error, omitting biased variables, solving heteroscedasticity, or serial correlation. FMOLS estimator according to Adom [58] is as follows:

$$\left[ \left( \sum_{t=1}^{T} (x_t - \overline{x}_i)(x_t - \overline{x})^T \right) \right]^{-1} * \left[ \sum_{t=1}^{T} (x_t - \overline{x}_i)\left( y_t^* + T\hat{\Delta}_{EM} \right) \right] \tag{9}$$

$$y_t^* = y_t - \hat{\Omega}_{EM} * \Omega_E^{-1} \Delta x_t$$

$$\widetilde{\Delta}_{EM} = \Delta_{EM} \hat{\Omega}_E^{-1} \Omega_{EM}$$

Here, $y_t^*$ is the transformed variable of the dependent variable to attain the endogeneity equation, and $\overset{\vee}{\Delta}_{EM}$ is the serial correlation correction.

DOLS is more useful for Johansen's cointegration, FMOLS, or CCR. DOLS is an extension of the Stock and Watson's [55] estimator, which is obtained from the following equation:

$$y_t = \alpha_i + \beta x_t + \sum_{j=1}^{k} c_j \Delta x_{t-j} + v_{it} \tag{10}$$

$c_j$ is the coefficient of a lead or lag being the first differenced explanatory variable. The estimated coefficient of DOLS is given by the following equation:

$$\hat{\beta}_{DOLS} = \left( \sum_{t=1}^{n} q_t q_t^T \right)^{-1} * \left[ \left( \sum_{t=1}^{n} q_t y_t^* \right)^{-1} \right] \tag{11}$$

Here, $q$ is $\{2(q + 1) \times 1\}$ vector of regressors and $q_t = x_t - \overline{x}_i \Delta x_{t-q}, \ldots\ldots\Delta x_{t+q}$.

Canonical cointegrated regression (CCR) introduced by Park [63] works like FMOLS. There is a difference between these two models. CCR works on data transformation only, while FMOLS converts data and parameters [58,63]. In addition, CCR is also a single equation-based regression, which can also be applied to multivariate regression. The CCR estimator is obtained as follows from Adom [58]:

$$\hat{\beta}_{CCR} = \left[ \left( \sum_{t=1}^{T} (x_t - \overline{x}_i) \right)(x_t - \overline{x})^T \right]^{-1} * \left[ \sum_{t=1}^{T} (x_t - \overline{x}_i) y_t^* \right] \tag{12}$$

## 5. Results and Discussions

### 5.1. Unit Root Test

In order to determine the stationarity properties of our series in reference to literature [56,57,59], a unit root test was conducted using ADF [46] and PP [47] unit root tests.

The results from these tests are presented in Table 2. In both cases, the study variables are stationary at the first difference.

**Table 2.** Unit root tests.

| | ADF Test | | PP Test | | Ng-Perron Test | |
|---|---|---|---|---|---|---|
| **Variable** | **Level** | **1st Diff.** | **Level** | **1st Diff.** | **Level** | **1st Diff.** |
| MIE | −0.056 | −3.215 ** | 0.289 | −2.143 ** | −5.234 *** | |
| SIE | −0.970 | −6.620 *** | −0.897 | −6.701 *** | −2.456 * | |
| WE | 0.245 | −3.205 ** | −1.005 | −4.123 ** | −8.776 *** | - |
| IFC | 0.062 | −8.567 *** | −1.987 | −5.777 *** | −4.132 *** | |
| GMI | 2.165 | −3.456 *** | −0.321 | −9.767 *** | −7.654 * | |
| GSI | −2.749 ** | - | 0.1567 | −2.654 *** | −5.460 *** | |

\*\*\*, \*\*, \* denotes 1%, 5%, and 10% significant level, respectively. MIE = manufacturing industry employment, SIE = service industry employment, WE = wage earning, IFC = import from China, GMI = growth in manufacturing industry, GSI = growth in service industry.

Further stationarity test was performed using NG-Perron test with the aim of showing the effectiveness of Ng-Perron in addressing the shortcomings of ADF test in reference to small sample size as opined in the literature [64,65]. The result from Ng-Perron as depicted in Table 2 reveals that only two of the variables (SIE and GIM) are weakly stationary at level, but became significant at the first difference. This result complements the results earlier obtained from ADF and PP unit root tests and established that the variables in this study are all stationary.

*5.2. Cointegration Analysis Result*

This study employed the Fisher's statistics to examine the underlying cointegration of the models in this study. The integrated cointegration measures includes the EG-JOH and EG-BO-BDM measures as shown in Table 3a–c. From the results presented in the table for the three (3) models, the EG-JOH and EG-JOH-BO-BDM's F-values are greater than a critical value and this implies that the null hypothesis of no cointegration can be rejected. In summary, a long-run cointegration between manufacturing employment (MIE), importation exposure from China (IFC), growth in manufacturing sector and service sector; service industry employment, importation exposure from China (IFC), growth in manufacturing sector and service sector; and wage earning, importation exposure from China (IFC), growth in manufacturing sector and service sector in the context of South Africa for the period 1992–2020. The recent cointegration test developed by Bayer and Hanck [53], which combined a cointegration approach to strengthen the cointegration analysis, was employed in this study and the results are presented in Table 3a–c. This innovative technique amalgamates findings of previous cointegration and shows on the Fisher F-statistic with consistent and decisive results. However, strong conviction is required for this hypothesis-based cointegration. That is, the order of each variable is stationary at first difference. An F-value greater than a critical indicates that the null hypothesis of no cointegration should be rejected; otherwise, an F-value less than the critical value accepts the null hypothesis of no cointegration. A value of F-statistic under 1%, 5%, and 10% level of significance shows the statics are higher than the critical values, thus, there exists a cointegration among the variables.

**Table 3.** (**a**). Cointegration result of Bayer–Hanck cointegration (Model 1). (**b**). Cointegration result of Bayer–Hanck cointegration (Model 2). (**c**). Cointegration result of Bayer–Hanck cointegration (Model 3).

| (a) | | | |
|---|---|---|---|
| Test Statistic | EO-JOH | EO-JOH-BO-BDM | Decision |
| F-statistic | 57.431 | 158.904 | |
| Critical regions | | | |
| 1% | 13 | 18.701 | Cointegration exist |
| 5% | 11.218 | 16.321 | |
| 10% | 7.121 | 15.595 | |
| (b) | | | |
| F-statistic | 51.431 | 198.804 | |
| Critical regions | | | |
| 1% | 12.212 | 16.701 | Cointegration exist |
| 5% | 10.218 | 18.301 | |
| 10% | 13.121 | 16.595 | |
| (c) | | | |
| F-statistic | 42.431 | 108.814 | |
| Critical regions | | | |
| 1% | 11.213 | 16.721 | Cointegration exist |
| 5% | 13.118 | 17.321 | |
| 10% | 9.101 | 11.495 | |

## 6. Long-Run Effect Analysis

Having established the possible cointegration among the variables, we proceeded to investigate the long-run causal relationship between the manufacturing industry employment (*lnMIE*) and the independent variables (*lnIFC, lnGMI, lnGSI*) for Model 1; service industry employment (*lnSIE*) and independent variables (*lnIFC, lnGMI, lnGSI*) for Model 2; and wage earning (*lnWE*) and independent variables (*lnIFC, lnGMI, lnGSI*) for Model 3.

The models were estimated using ARDL estimator and the results are presented in Table 4. The ARDL estimator's results as presented in Table 4, column 1 for model 1 reveal a negative and significant coefficient of lnIFC (−0.097 ***), which implies that holding all other variables constant, a percentage change in import competition from China will reduce the manufacturing employment in South Africa's labour market in the long-run by 0.10%. The sign of the coefficient of ARDL is consistent with that of FMOLS, DOLS and CCR (see Tables 5–7), which provides the robustness check for ARDL estimator, and the three estimators suggest that the coefficients are significant at less than 1% significant level. Similarly, the growth in the service industry as a control variable was found to have a negative and significant long-run effect on the manufacturing industry employment. These estimates from ARDL are confirmed by the FMOLS, DOLS and CCR estimators and are significant at 5% significant level. Meanwhile, the growth in the manufacturing industry (lnGMI) was found to have a positive and significant long run effect on the manufacturing employment. The positive relationship demonstrated by ARDL estimator as depicted in Table 4 is consistent with the sign of coefficient from FMOLS, DOLS and CCR (see Tables 5 and 7), which are all significant at 5% significance level.

**Table 4.** ARDL long and short-run estimates.

| Variable | Model 1<br>*lnMIE = f(lnIFC, lnGMI, lnGSI)* | Model 2<br>*lnSIE = f(lnIFC, lnGMI, lnGSI)* | Model 3<br>*lnWE = f(lnIFC, lnGMI, lnGSI)* |
|---|---|---|---|
| Long-run estimates | | | |
| lnIFC | −0.097 *** | −0.999 *** | −7.18 ** |
| lnGMI | 3.36 ** | 0.004 *** | 0.011 ** |
| lnGSI | −20.80 ** | −0.126 ** | −0.391 |
| Short-run estimates | | | |
| ΔlnIFC | 0.045 | 0.000182 | −4.37 |
| ΔlnGMI | 1.24 | 0.871 | 0.039 |
| ΔlnGSI | 0.65 | 0.128 ** | −0.564 |
| $ECT_{t-1}$ | −0.032 *** | −0.00066 *** | −0.023 ** |
| Diagnostic tests | | | |
| LM test | 16.45 (0.09) | 13.35 (0.079) | 9.213 (0.24) |
| Heteroscedasticity test | 9.67 (0.345) | 3.933 (0.13) | 7.66 (0.103) |
| Normality test | 0.67 (0.12) | 0.79 (0.71) | 0.437 (0.07) |
| Stability test | Yes | Yes | Yes |

Note: ***, ** denotes 1% and 5% significance level.

**Table 5.** Long run effect of lnIFC, lnGMI and lnGSI on lnMIE (Model 1).

| | FMOLS | | DOLS | | CCR | |
|---|---|---|---|---|---|---|
| Variable | Coefficient | *p* Value | Coefficient | *p* Value | Coefficient | *p* Value |
| lnIFC | −0.639 | 0.000 | −0.03 | 0.000 | −1.010 | 0.000 |
| lnGMI | 0.630 | 0.001 | 0.79 | 0.000 | 1.131 | 0.000 |
| lnGSI | −0.018 | 0.020 | −0.08 | 0.032 | −0.070 | 0.000 |
| Constant | 1.221 | 0.102 | 0.49 | 0.010 | 1.911 | 0.231 |

**Table 6.** Long run effect of lnIFC, lnGMI and lnGSI on lnSIE (Model 2).

| | FMOLS | | DOLS | | CCR | |
|---|---|---|---|---|---|---|
| Variable | Coefficient | *p* Value | Coefficient | *p* Value | Coefficient | *p* Value |
| lnIFC | 0.03 | 0.000 | 0.566 | 0.000 | 0.790 | 0.000 |
| lnGMI | −0.79 | 0.001 | −0.560 | 0.000 | −0.343 | 0.000 |
| lnGSI | 0.08 | 0.020 | 0.023 | 0.032 | 0.212 | 0.000 |
| Constant | 0.49 | 0.102 | 1.134 | 0.010 | 1.651 | 0.231 |

**Table 7.** Long run effect of lnIFC, lnGMI and lnGSI on lnWE (Model 3).

| | FMOLS | | DOLS | | CCR | |
|---|---|---|---|---|---|---|
| Variable | Coefficient | *p* Value | Coefficient | *p* Value | Coefficient | *p* Value |
| lnIFC | −0.163 | 0.000 | −0.071 | 0.010 | −0.023 | 0.001 |
| lnGMI | 0.179 | 0.000 | 0.432 | 0.020 | −0.669 | 0.010 |
| lnGSI | 0.208 | 0.420 | 0.710 | 0.062 | 0.018 | 0.080 |
| Constant | 0.349 | 0.102 | 0.003 | 0.010 | 0.429 | 0.231 |



The significant negative long-run effect of imports from China on the manufacturing industry demonstrated in this study is congruent with some previous studies that concluded that international trade has a significant effect on the local labour market [2–4,12–15,19,21,23]. In addition, the studies of Balsvik et al. [26] and Utar [42] demonstrated a negative influence of Chinese imports on manufacturing employment in the Norway and Danish textile and clothing industries, respectively, was corroborated our findings, which suggest a negative long-run effect of Chinese imports on South African manufacturing industry employment. The exogenous shock like international trade through the channel of the import-demand shock as opined by Autor et al. [25,38] was confirmed in our study, which reveals a significant influence of importation shock from China on the manufacturing and service employment in the context of South Africa. Moreover, the expectation of a large negative impact like unemployment on regional labour market, especially those concentrated in manufacturing industry as estimated by Jiang et al. [11], was also demonstrated in this study, which found a negative significant long-run relationship of importation shock from China on the manufacturing employment in South Africa. The finding from our study also corroborates the position of the studies of Donoso et al. [39] and Bilici [40], who conducted similar studies in Spain and France, respectively, and concluded that the Chinese import competition mostly impacted on the manufacturing sectors' labour force. Meanwhile, our finding contradicts the position of Dauth et al. [41] and Jiang et al. [11], who did similar studies, but could not establish the significant effect of Chinese imports on the labour market in the European countries and Sweden, respectively.

In model 2 of this study, we examined the long run effect of China imports (lnIFC), growth in manufacturing industry (lnGMI) and growth in services industry (lnGSI) on the service industry employment (lnSIE). The results from the ARDL estimator (see Table 4), which was confirmed with the robustness estimates from FMOLS, DOLS and CCR as presented in Tables 5–7, shows that while China's imports and growth in the service industry have a positive and significant long-run effect on service industry employment, the growth in manufacturing industry was found to have a negative and significant long-run effect on the service industry employment. This is an indication that in the context of South Africa, a change in the manufacturing industry contributes to the drifting of employees away from the service industry. The positive and significant long-run effect of China's imports on service industry employment demonstrated in this study contradicts the position of Jiang et al. [11], who concluded in their study conducted in the context of Sweden's labour market that Chinese imports have no significant effect on the non-manufacturing industry of Sweden's labour market.

In reference to the possible influence of Chinese imports on the workers' earnings as suggested in the literature, this study examined the long-run effect of China's imports on wage earnings in the South African labour market. The investigation was conducted using ARDL and FMOLS, DOLS and CCR as a robustness check. From the result presented in Table 4, it shows that China's imports have a negative and significant long-run effect on the workers' earnings (−7.18 **) in the context of South Africa (see Table 4), and this is consistent with the three estimators, which also established the coefficient of IFC to be negative and significant at less than 5% significance level. However, while growth in the manufacturing industry was revealed to have a positive and significant long-run effect on workers' earnings (0.011 **), the growth in the service industry shows no significant relationship with workers' earnings. The negative and significant long-run effect of China's imports on workers' earnings demonstrated in this study implies that a change in China's imports will have a significant effect on the workers' earnings in South Africa in the long-run. The negative and significant long-run effect of China's imports on the labour market outcome like wage is in congruence with some studies [3–5,11], which concluded that China's imports have a significant effect on the workers' earnings. Meanwhile, the finding contradicts the studies of Balsvik et al. [26], who could not establish a significant effect of China's imports on workers' earnings, and Citino and Linarello [15], who found a positive effect of China's imports on wages in their study.

Moreover, the outcome of the ECT for the three models as presented in Table 4, which shows to be −0.32 ***, −0.00066 ***, and −0.023 *** for Model 1, Model 2, and Model 3, respectively, indicates that any shock in the model is adjusted and corrected at the speed of 32%, 0.06%, and 2.3%. In addition, the negative and significance of the ECT coefficient confirms the stable long-run relationship between the dependent variable and independent variables in each of the models. In addition, the outcomes of the diagnostic tests are presented in Table 4, and these tests affirm the absence of autocorrelation, reveal the normal distribution of the model, as well the stability of the models, which are depicted in Figures 4–6 for Models 1–3 CUSUM and CUSUM of square, respectively.

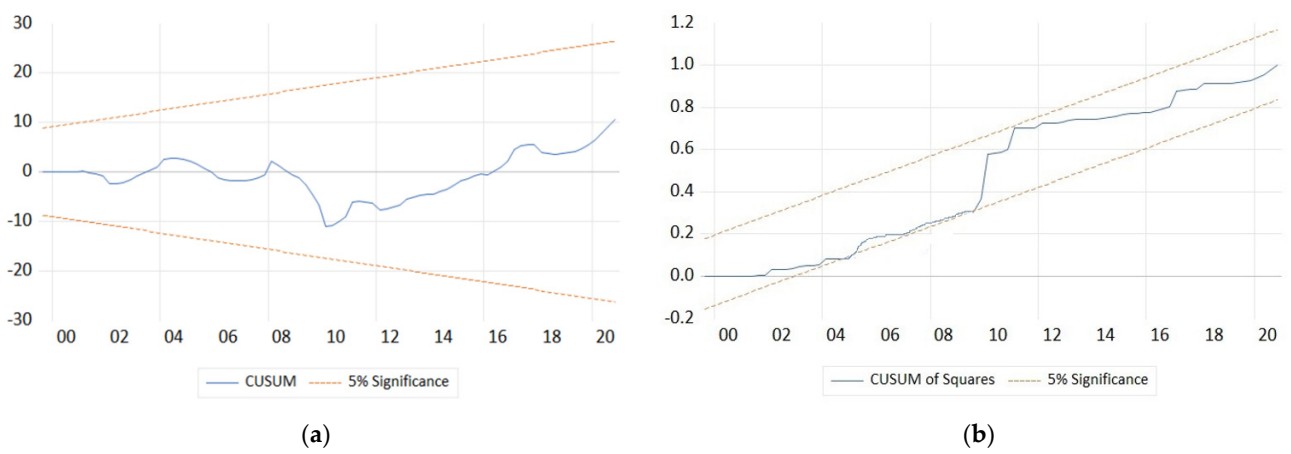

**Figure 4.** (**a**). Stability test (Model 1). (**b**). Stability test (Model 1).

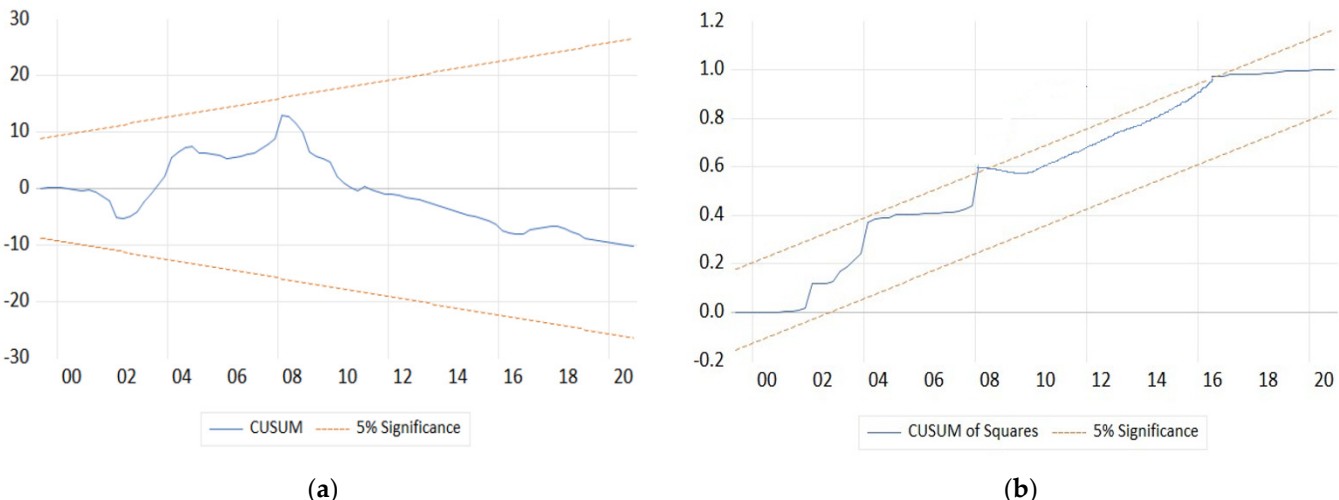

**Figure 5.** (**a**). Stability test (Model 2). (**b**). Stability test (Model 2).

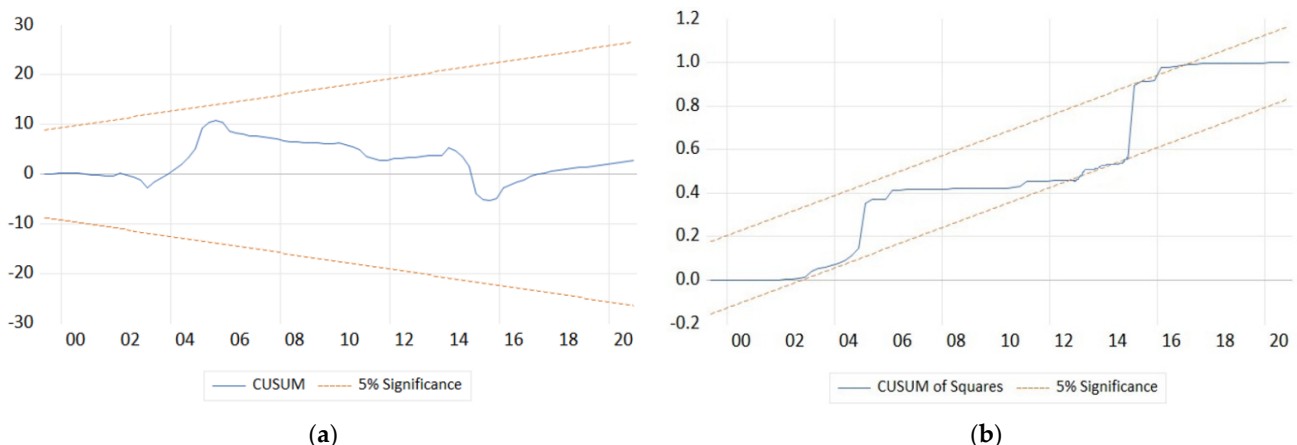

**Figure 6.** (**a**). Stability test (Model 3). (**b**). Stability test (Model 3).

## 7. Conclusions

The continuous exposure of South Africa to China's import competition is similar to the concern raised in the literature on the possible consequence on the labour market. Many previous studies have examined the impact of imports or international trade on either the manufacturing industry or employees of the manufacturing sector. Meanwhile, most of these studies are tilted towards the developed countries with little or none that examine the implication of exposure of developing countries to China's imports on labour market outcomes, whereas these developing countries are at a developing stage that requires the contribution of their industries for the sustainability of their growth. Thus, our paper investigates the effect of exposure of South Africa to China's imports on its labour market. Specifically, we investigate the following hypotheses: whether imports from China influence (i) the manufacturing industry employment, (ii) non-manufacturing industry employment, (iii) wage earning of employees in South Africa, and (iv) to test the hypotheses of whether growth in the manufacturing and non-manufacturing industry influences the manufacturing and non-manufacturing industry employment, as well as wage earning in South Africa's labour market.

In order to address the study objective, we utilized a South Africa yearly data from 1992 to 2020, which was converted to quarterly data to estimate our models. This study controls for the determinants such as import competition from China, the growth in manufacturing and non-manufacturing industry. Our analysis finds that exposure to Chinese competition of South Africa exerts a negative effect on the local labour market outcomes, which is not only on manufacturing sector employment and workers earning, but also triggers a positive effect on the non-manufacturing industry employment. These findings imply that the hypotheses stated in this study were valid. These findings are also congruent with the findings of Blanco et al. [12], Blyde et al. [13], and Citino and Linarella [15], and support the interpretation proposed by Autor et al. [25,38]. These findings indicate that the reduction in the manufacturing industry employment levels results in a decline in the average earnings of South African households. The implication of Chinese trade for South African employment, household earning may trigger public ambivalence towards globalization and specific anxiety about increasing trade with China. This is expected owing to the fact that the manufacturing industry in South Africa and Africa in general is still low, and there are limited manufactured goods from locally produced manufacturers that are intended for domestic consumptions. Meanwhile, the South African manufacturing industry would possibly be at a great disadvantage in their domestic markets as a result of the competition from China. In these regards, the policy makers in South Africa should play a decisive role in ensuring the development of the manufacturing sector to ensure the sustainability of their labour market. Nevertheless, our findings put the current discussion about import competition especially from China into perspective in the context of a developing country.

The limits of the research derive primarily from the limitation imposed by the variables used. For this reason, further research is a necessity. Future Chinese import competition research can be conducted on two levels. One of the directions may be to consider adding more variables for the same South Africa. For example, the control of high-skilled employees and low-skilled employees in different industries. Additionally, the control of immigration to ascertain the possible effect of these on the labour market. A second direction, we envisage conducting a similar study selected African countries as a panel study. These countries have certain characteristics generated by belonging to the same continent and on almost the same economic growth level. For these countries, the understanding of the effect of their exposure to competition of China import will enable their policy makers to put in place the measure that would protect the labour market from the Chinese imports.

**Author Contributions:** The article was conceptualized and the methodology was designed by both M.F. and M.B., Data collection and analysis were done by M.F., while the results validation was carried out by M.B. The manuscript was drafted by M.F. with the support of M.B. for the editing and proofreading. All authors have read and agreed to the published version of the manuscript.

**Funding:** This research received no external funding.

**Conflicts of Interest:** The authors declare no conflict of interest.

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
