# Peer review of "Sustainability of Local Labour Market in South Africa: The Implications of Imports Competition from China"

_sustainability, doi:10.3390/su14127168_

Round 1
Reviewer 1 Report
I do not think the revision clearly addressed the issues pointed by the reviewers.
Author Response
We appreciate the insightful comment of the anonymous reviewer which has assisted us in improving the standard of the manuscript. We took our time to respond appropriately to every issue raised by the reviewer, and the evident in the comments by the two anonymous reviewers.
Reviewer 2 Report
This version of the paper is a significant improvement over the previous version and is ready for publication.
Figure 4 to Figure 6 are not very clear, it is recommended to improve it.
Author Response
We appreciate the anonymous review for the positive comment.
We appreciate the anonymous reviewer for the insightful observation. The authors have improved the figures as suggested (see pages 19-21)
Reviewer 3 Report
Dear authors, congratulations for the version of this article, which is very well done.
I suggest that it be accepted for publication and I wish you good luck for future articles.
Best Regards,
Author Response
We appreciate the anonymous review for the positive comment.
This manuscript is a resubmission of an earlier submission. The following is a list of the peer review reports and author responses from that submission.
Round 1
Reviewer 1 Report
The data and variables are very unclearly and poorly explained, so the readers really cannot what was analysed and how. A rough description of data sources is not enough. you have to explain each variable, how those variables are measured and when and you also have to show the summary statistics.
Reviewer 2 Report
The paper examined the impact of import from China, growth in manufacturing and non-manufacturing industry on by using South African manufacturing and non-manufacturing industry employment, import from China, growth in manufacturing and non-manufacturing industry, and workers' earning data, manufacturing and non-manufacturing industry employment and workers' earning. It seems to have a certain novelty. However, the paper has obvious shortcomings.Overall, the paper is more like a graduate course paper than an academic paper that can be published in an academic journal. The reasons are as follows:
First, the analysis method (FMOLS, DOLS, and CRR) used in this study was very cutting-edge 20 years ago, but it is now a bit old.Students who have studied time series analysis today basically can do these models and analyses.
Second, from the perspective of time series analysis, it is really impossible to analyze such a macroeconomic phenomenon with only 28 years of time series data.Therefore, there are great doubts about the reliability of the results obtained in the present paper.
Third, in the literature review part, there is no clear hypothesis that can be tested, which is a non-standard practice.
It can be seen from the above points that there is still much room for improvement in the academic skills of the authors of this paper.
Reviewer 3 Report
Dear authors,
The article is well structured, well presented, with clear ideas, with a correct analysis and interpretation of the data and with conclusions in line with all the discussion that precedes it and with the objectives initially defined.
In the introduction, it is possible to perceive the scarcity of studies focused on the objectives that the authors present. However, the authors mention studies that allow a confrontation of ideas and the gap is well defined.
The results and their discussion are in line with the objectives of the study.
Points to improve:
A spelling check is required – some words are misspelled (eg. obesrved – line 47; espoused – line 40; outcomess – line 36);
A review of the sentence construction is necessary, where, for example, repetitions of ideas are found (These findings indicate that the reduction in reduction in the manufacturing industry levels results… - line 449/450
Good luck